# *Mycobacterium ulcerans* dynamics in aquatic ecosystems are driven by a complex interplay of abiotic and biotic factors

Andrés Garchitorena[1,2]*, Jean-François Guégan[1,2,6], Lucas Léger[1], Sara Eyangoh[3], Laurent Marsollier[4], Benjamin Roche[5]

[1]Maladies Infectieuses et Vecteurs: Ecologie, Génétique, Evolution et Contrôle (UMR CNRS/IRD/UM 5290), Montpellier, France; [2]Ecole des Hautes Etudes en Santé Publique, Rennes, France; [3]Laboratoire de Mycobactériologie, Centre Pasteur du Cameroun, Réseau International des Instituts Pasteur, Yaoundé, Cameroon; [4]Institut National de la Recherche Médicale U892 (INSERM) et CNRS U6299, équipe 7, Université et CHU d'Angers, Angers, France; [5]International Center for Mathematical and Computational Modelling of Complex Systems (UMI IRD/UPMC UMMISCO), Bondy Cedex, France; [6]International programme Future Earth, ecoHEALTH Initiative, Ottawa, Canada

**Abstract** Host–parasite interactions are often embedded within complex host communities and can be influenced by a variety of environmental factors, such as seasonal variations in climate or abiotic conditions in water and soil, which confounds our understanding of the main drivers of many multi-host pathogens. Here, we take advantage of a combination of large environmental data sets on *Mycobacterium ulcerans* (*MU*), an environmentally persistent microorganism associated to freshwater ecosystems and present in a large variety of aquatic hosts, to characterize abiotic and biotic factors driving the dynamics of this pathogen in two regions of Cameroon. We find that *MU* dynamics are largely driven by seasonal climatic factors and certain physico-chemical conditions in stagnant and slow-flowing ecosystems, with an important role of *pH* as limiting factor. Furthermore, water conditions can modify the effect of abundance and diversity of aquatic organisms on *MU* dynamics, which suggests a different contribution of two *MU* transmission routes for aquatic hosts (trophic vs environmental transmission) depending on local abiotic factors.

*For correspondence: andres.garchitorena@gmail.com

**Reviewing editor**: Quarraisha Abdool Karim, University of KwaZulu Natal, South Africa

## Introduction

Despite increased understanding of infectious disease ecology and dynamics, recent decades have seen an upsurge in the emergence and re-emergence of multiple infectious diseases (*Jones et al., 2008*). Most emerging pathogens are zoonotic and have a very broad range of hosts (*Woolhouse et al., 2001*; *Woolhouse and Gowtage-Sequeria, 2005*), and as a result, host–parasite interactions are often embedded within complex host communities (*Plowright et al., 2008*; *Roche et al., 2013b*). Biotic interactions between hosts in the community may play an important role on disease transmission, either promoting or diluting the overall prevalence of the pathogen (*LoGiudice et al., 2003*; *Keesing et al., 2010*; *Johnson et al., 2013*; *Roche et al., 2013a*). In addition, the effect of seasonal variations in climate on abiotic conditions in water and soil can influence the composition of host communities and in turn have an impact on the ecological dynamics of the pathogens

**eLife digest** *Mycobacterium ulcerans* is a slow-growing bacterium that causes a rare tropical disease in humans called Buruli ulcer. The infection affects the skin and underlying tissues, initially causing small painless lesions that can develop into large open sores or ulcers that are responsible for significant handicap in rural areas of sub-Saharan Africa.

Disease outbreaks generally occur in close association with stagnant and slow-flowing aquatic ecosystems, mostly after floods or other large environmental disturbances. It is believed that contact with water sources that contain the bacteria causes infection in humans, but the specific mode of transmission remains a mystery. By defining the factors that influence the bacteria's presence in the environment, public health officials could develop initiatives that would reduce an individual's risk of infection when conditions support a *M. ulcerans* outbreak.

To identify the environmental conditions that affect the prevalence of *M. ulcerans* in two regions of Cameroon where Buruli ulcer is present, Garchitorena et al. have now analyzed a large amount of ecological data about the bacteria using cutting-edge statistical techniques. This revealed that the amount of *M. ulcerans* varies following seasonal changes in climate, at least in the region dominated by tropical rainforest. In this region, the bacteria are also generally present in waters that are more alkaline and contain fewer animals, especially from certain species that could prevent the infection spreading to other aquatic hosts. In the other region, dominated by a savannah landscape, the bacteria are most abundant in stagnant or slowly moving waters that have optimal physical and chemical conditions and contain many diverse species of potential animal hosts. The discovery of contrasting results for the two regions suggests that there are at least two ways that *M. ulcerans* can persist in the environment and infect the aquatic animals. The prevailing method—through environmental transmission or through interactions between hosts—depends on the properties of the water.

Many other infectious diseases are caused by pathogens that, like *M. ulcerans*, infect many different hosts and persist in the environment for long periods. Future research following methods like those used by Garchitorena et al. would help to reveal whether these pathogens are affected by environmental factors in similar ways to *M. ulcerans*.

(*Ostfeld et al., 2008*). Because of the intertwined nature of biotic and abiotic drivers, disentangling their respective contribution to pathogen dynamics and transmission through complex and different-scale processes remains a challenge (*Plowright et al., 2008*). Nonetheless, identifying the underlying ecological mechanisms driving the emergence and persistence of diseases is essential to reduce disease risk in human populations (*Roche and Guégan, 2011*).

An illustrative example is the case of *Mycobacterium ulcerans* (*MU*), an environmentally persistent microorganism associated to freshwater ecosystems in tropical countries and present in a large variety of aquatic hosts (*Benbow et al., 2008*; *Marion et al., 2010*; *Garchitorena et al., 2014*). From a public health perspective, *MU* is the agent responsible for Buruli ulcer (BU), a devastating skin disease with great health and socio-economic consequences in tropical and subtropical countries (*WHO, 2008*). Emergence, distribution, and risk factors for BU in many parts of the world are associated with stagnant and slow-flowing ecosystems (*Brou et al., 2008*; *Wagner et al., 2008*; *Jacobsen and Padgett, 2010*; *Marion et al., 2011*). The environmental factors that favour *MU* persistence and transmission within these ecosystems are still poorly understood but the environmental and multi-host nature of the pathogen suggests that its environmental dynamics can be the result of a complex interplay between environmental factors and biotic interactions (*Garchitorena et al., 2014*; *Morris et al., 2014*).

*MU* is broadly present across taxa in aquatic communities over space and time (*Benbow et al., 2008*; *Marion et al., 2010*; *Garchitorena et al., 2014*), suggesting that a multiplicity of hosts can play a role in *MU* persistence in the environment. Biotic interactions between hosts are thought to be a pathogen transmission route between organisms (*Merritt et al., 2010*), with *MU* being integrated in the aquatic community from the environment thanks to filter feeder, herbivorous, and scavenger organisms and then transmitted across the community through predation (*Marsollier et al., 2002*, *2007a*, *2007b*; *Mosi et al., 2008*). As a result, community-level factors such as biodiversity or

abundance of aquatic organisms could drive the environmental load of *MU* through amplification or dilution effects, as demonstrated for other pathogens (*Ezenwa et al., 2006*; *Suzán et al., 2009*; *Keesing et al., 2010*; *Johnson et al., 2013*). Furthermore, some keystone species could play an overwhelming role in the transmission and overall *MU* prevalence in host communities (*Roche et al., 2013a*).

Some specific water conditions, physical or chemical, could also favour *MU* environmental persistence and dynamics in aquatic ecosystems. *MU* seems to grow better under laboratory conditions with low oxygen, high temperature, and mildly acidic *pH* (*Portaels and Pattyn, 1982*; *Palomino and Portaels, 1998*; *Palomino et al., 1998*; *Dega et al., 2000*; *Marsollier et al., 2004*). Genomic studies show that *MU* is sensitive to UV light (*Stinear et al., 2007*; *Doig et al., 2012*) so turbid or protected environments could promote *MU* persistence. Many of these conditions are generally met in swamps and other stagnant and slow-flowing ecosystems, and therefore, if optimal abiotic conditions are met, *MU* could grow and persist in these ecosystems as free-living stages in the water and infect aquatic organisms directly, without the need for a trophic transmission to take place. Besides, numerous abiotic factors within aquatic ecosystems can influence host community structures and assemblages (*Eric Benbow et al., 2013*; *Garchitorena et al., 2014*), since aquatic invertebrate and vertebrate taxa have different ranges of optimal water conditions (*Dickens and Graham, 2002*). This could represent an indirect influence of abiotic conditions on *MU* transmission within aquatic communities.

A direct transmission of *MU* driven mostly by abiotic factors and a trophic transmission driven by biological interactions are not two mutually exclusive routes but rather could complement each other to allow persistence of *MU* under a wide range of environments. Identifying the contribution of such transmission routes requires a deep understanding of the dynamics of *MU* within aquatic communities through space and time. A recent characterization of *MU* dynamics with unprecedented detail in two BU endemic areas with very distinct environmental conditions (*Garchitorena et al., 2014*) offers the variability needed to address this question. Indeed, in Bankim, a region located in a transition zone between forest and savannah, swamps had remarkably higher *MU* positivity, as initially expected, whereas in Akonolinga, where rainforest is the prevailing landscape, all ecosystems had similar *MU* positivity. These regional differences suggested that savannah swamps had unique favourable conditions for *MU* that were not found elsewhere, but *MU* was still able to persist in unfavourable environments. Furthermore, temporal fluctuations in *MU* presence in Akonolinga suggested a potential role of seasonal climatic events as drivers of *MU* dynamics.

Relying on this work, the aim of this paper is to study for the first time the contribution of ecological factors, both biotic and abiotic, to the dynamics of *MU* in the aquatic environment. More specifically, we attempt to identify a set of abiotic conditions that could be optimal for *MU* growth and allow direct transmission to aquatic organisms, likely in stagnant waters and, in the absence of these, explore which biotic factors could still allow *MU* to persist, potentially through trophic transmission. Insights into the ecological mechanisms allowing for *MU* growth and persistence over space and time, while accounting for the potential impact of seasonal climatic events, may have profound implications for understanding BU risk to human populations.

To address these questions, we model *MU* positivity in 32 aquatic communities over time with generalized linear mixed models (GLMMs), including all relevant seasonal, abiotic, and biotic factors as fixed effects. We use cutting-edge multi-model selection procedures and information theory to identify and quantify the most important predictors of *MU* dynamics, using a genetic algorithm to screen multiple models from all potential combinations of explanatory variables and making inference from a set of weighted best-performing models. In addition, we back the results of this novel approach, which deals with the uncertainty associated with model selection, by comparing them to those obtained by classical model selection procedures. We then discuss the implications of disentangling biotic and abiotic factors for host/parasite interactions and the importance of rigorously analysing the underlying drivers of pathogen dynamics mediated through complex and different-scale processes.

## Results

By simultaneously accounting for multiple seasonal, abiotic, and biotic factors, our results show the complex interplay that drives *MU* environmental dynamics. In a previous step to the statistical modelling of *MU* positivity, we performed principal component analysis (PCA) of the most relevant

physico-chemical characteristics in the water, as a means to explore common patterns in the ecosystems and to include these PCs as alternative abiotic predictors in the model. The ecosystems sampled and followed up in both Akonolinga and Bankim regions consistently revealed common physico-chemical patterns depending on water flow (*Table 1*). The first PC, explaining about 50% of variation in aquatic ecosystems in both regions, showed a positive correlation between water flow, dissolved oxygen, and *pH*, while temperature was inversely correlated. Furthermore, the inverse correlation between temperature and water flow held in the second PC, and the positive correlation between water flow and oxygen was also present in the third PC. Using multi-model selection for our GLMMs, a combination of seasonal factors, water conditions (abiotic factors), and community-level characteristics (biotic factors) remained as important predictors of *MU* positivity in the final set of best models for Akonolinga (*Table 2*) and Bankim (*Table 3*). Among all possible models tested, 39 were selected for estimation of model average estimates in Akonolinga and 100 in Bankim, since these models were similarly performing according to their Akaike Information Criterion (AIC) scores (see the methodology section). Although all factors were included together as part of the full model, the results explained below are divided in groups of factors for clarity. Furthermore, comparison of multi-model inference results with those obtained by classical model selection procedures can be found in Appendix 1, section 1.

## Effect of seasonality on *M. ulcerans*

Seasonality was investigated by including *sine* and *cosine* functions as independent predictors of *MU* positivity. The presence of *MU* in aquatic ecosystems in Akonolinga was associated to seasonal variations with a single annual cycle as revealed by the positive effect of the *sine* function on *MU* ($b = 0.36$; 95% confidence interval (CI) [0.04, 0.67], $w_i = 1$) and the presence of this variable in all best models (*Table 2*). On the contrary, the seasonal effect in Bankim was not apparent, where only 4 months of collection were available, and none of the *sine* and *cosine* functions were important in the final models for this region (*Table 3*).

## Effect of abiotic conditions on *M. ulcerans*

Among all abiotic conditions, included in the models both as individual physico-chemical variables or through their combined effect as PCs, *pH* had a significant positive effect in all best models in Akonolinga (*Table 2*), either through the effect of component 2 that is directly correlated with *pH* ($b = 0.49$; 95% CI [0.21, 0.76], $w_i = 0.56$) or as individual variable in the remaining models ($b = 7.15$; 95% CI [2.56, 11.73], $w_i = 0.44$). In Bankim, however, the most important abiotic factor was water flow (*Table 3*). Lentic water bodies (low water flow) had significantly lower *MU* positivity than stagnant waters ($b = -1.80$; 95%CI [−3.04, −0.56], $w_i = 1$), and lotic water bodies (high water flow) had the lowest *MU* positivity ($b = -3.63$; 95% CI [−5.35, −1.91], $w_i = 1$). It is important to note that water flow in most environments was directly correlated with *pH*, and thus each region provides contrasting results for this abiotic factor.

**Table 1**. Description of environments defined by principal components analysis (PCA) of physico-chemical parameters

| | Akonolinga | | | | Bankim | | | |
|---|---|---|---|---|---|---|---|---|
| | PC1 | PC2 | PC3 | PC4 | PC1 | PC2 | PC3 | PC4 |
| Variance explained | 0.47 | 0.31 | 0.14 | 0.09 | 0.59 | 0.22 | 0.13 | 0.07 |
| Loadings | | | | | | | | |
| *pH* | −0.33 | 0.7 | −0.47 | 0.41 | −0.51 | 0.51 | −0.43 | 0.55 |
| Dissolved oxygen | −0.6 | 0.31 | 0.31 | −0.67 | −0.57 | 0.33 | 0.11 | −0.75 |
| Water flow | −0.59 | −0.32 | 0.46 | 0.59 | −0.51 | −0.29 | 0.72 | 0.36 |
| Temperature | 0.43 | 0.55 | 0.69 | 0.19 | 0.4 | 0.74 | 0.53 | 0.1 |

Separate PCA was performed for Akonolinga and Bankim, and only the most potentially relevant parameters were included.

**Table 2**. Results from multi-model selection for Akonolinga (12 months of sampling)

| Variable | Avg. effect (b) | Uncond. SE | Lower CL | Upper CL | Relative importance (w_i) | Nb. models |
|---|---|---|---|---|---|---|
| **(Intercept)** | 2.71 | 4.04 | −5.21 | 10.63 | 1.00 | 39 |
| Seasonality | | | | | | |
| **Sine (2pi*Month/12)** | **0.36** | **0.16** | **0.04** | **0.67** | **1.00** | **39** |
| Sine (2pi*Month/4) | – | – | – | – | – | – |
| Cosine (2pi*Month/12) | – | – | – | – | – | – |
| Cosine (2pi*Month/4) | – | – | – | – | – | – |
| Physico-chemical parameters | | | | | | |
| *pH* | **7.15** | **2.34** | **2.56** | **11.73** | **0.44** | **17** |
| Flow | – | – | – | – | – | – |
| Temperature | – | – | – | – | – | – |
| Dissolved oxygen | – | – | – | – | – | – |
| Conductivity | – | – | – | – | – | – |
| Iron | – | – | – | – | – | – |
| Physico-chemical parameters (PCA) | | | | | | |
| **PC2** | **0.49** | **0.14** | **0.21** | **0.76** | **0.56** | **22** |
| PC1 | – | – | – | – | – | – |
| PC3 | – | – | – | – | – | – |
| Community | | | | | | |
| **Abundance** | **−0.71** | **0.18** | **−1.07** | **−0.35** | **1.00** | **39** |
| Shannon | – | – | – | – | – | – |
| Aquatic taxa (%) | | | | | | |
| **Gastropoda** | **−0.58** | **0.17** | **−0.92** | **−0.24** | **1.00** | **39** |
| Oligochaeta (Presence) | 0.40 | 0.28 | −0.14 | 0.95 | 0.92 | 36 |
| Odonata | 0.08 | 0.15 | −0.21 | 0.37 | 0.87 | 34 |
| Hydracarina | 0.19 | 0.29 | −0.39 | 0.76 | 0.85 | 33 |
| Trichoptera | −0.01 | 0.16 | −0.31 | 0.30 | 0.67 | 26 |
| **Decapoda (Presence)** | **−1.10** | **0.38** | **−1.84** | **−0.35** | **0.59** | **23** |
| Hirudinea (Presence) | 0.48 | 0.26 | −0.02 | 0.99 | 0.59 | 23 |
| Coleoptera | 0.24 | 0.20 | −0.15 | 0.63 | 0.54 | 21 |
| **Hemiptera** | **−0.54** | **0.21** | **−0.94** | **−0.13** | **0.54** | **21** |
| **Anura** | **−0.41** | **0.16** | **−0.73** | **−0.09** | **0.41** | **16** |
| Ephemeroptera | 0.07 | 0.15 | −0.21 | 0.36 | 0.21 | 8 |
| Diptera | −0.07 | 0.15 | −0.38 | 0.23 | 0.10 | 4 |

Variables within each category are ordered by their relative importance. Variables with their 95% confidence interval (CI) with the same sign are represented in bold. Rare aquatic taxa are introduced in the model as Presence/Absence variables, while relative abundance is used for more abundant taxa.

## Effect of biotic interactions on *M. ulcerans*

The impact of aquatic communities on *MU* was studied through the effect of both individual aquatic taxa and community-level factors such as abundance and diversity. In Akonolinga, we found a negative association between total abundance and *MU* presence in all final models ($b = -0.71$; 95% CI [−1.07, −0.35], $w_i = 1$) and individual effects of several taxa. Individual taxa inversely correlated with *MU* were Gastropoda in all models ($b = -0.58$; 95% CI [−0.92, −0.24], $w_i = 1$) and Decapoda ($b = -1.10$; 95% CI [−1.84, −0.35], $w_i = 0.59$), Hemiptera ($b = -0.54$; 95% CI [−0.94, −0.13], $w_i = 0.54$), and Anura ($b = -0.41$, 95% CI [−0.73, −0.09], $w_i = 0.41$), with lower importance in the final models

**Table 3**. Results from multi-model selection for Bankim (4 months of sampling)

| Variable | Avg. effect ($b$) | Uncond. SE | Lower CL | Upper CL | Relative importance ($w_i$) | Nb. Models |
|---|---|---|---|---|---|---|
| **(Intercept)** | **−13.98** | **4.13** | **−22.07** | **−5.89** | **1.00** | **100** |
| Seasonality | | | | | | |
| Sine (2pi*Month/12) | – | – | – | – | – | – |
| Sine (2pi*Month/4) | – | – | – | – | – | – |
| Cosine (2pi*Month/12) | – | – | – | – | – | – |
| Cosine (2pi*Month/4) | – | – | – | – | – | – |
| Physico-chemical parameters | | | | | | |
| **Water flow (lentic)** | **−1.80** | **0.63** | **−3.04** | **−0.56** | **1.00** | **100** |
| **Water flow (lotic)** | **−3.63** | **0.88** | **−5.35** | **−1.91** | **1.00** | **100** |
| *pH* | – | – | – | – | – | – |
| Temperature | – | – | – | – | – | – |
| Dissolved oxygen | – | – | – | – | – | – |
| Conductivity | – | – | – | – | – | – |
| Iron | – | – | – | – | – | – |
| Physico-chemical parameters (PCA) | | | | | | |
| PC3 | 0.44 | 0.39 | −0.33 | 1.21 | 0.07 | 7 |
| PC2 | 0.34 | 0.41 | −0.47 | 1.15 | 0.03 | 3 |
| **PC1** | **0.67** | **0.33** | **0.03** | **1.32** | **0.01** | **1** |
| Community | | | | | | |
| Abundance | 0.86 | 0.47 | −0.06 | 1.79 | 1.00 | 100 |
| **Shannon** | **4.29** | **1.21** | **1.93** | **6.66** | **1.00** | **100** |
| Aquatic taxa (%) | | | | | | |
| Gastropoda | −0.39 | 0.30 | −0.97 | 0.18 | 0.90 | 90 |
| Anura | −0.54 | 0.37 | −1.26 | 0.19 | 0.89 | 89 |
| Trichoptera | −0.05 | 0.64 | −1.30 | 1.20 | 0.89 | 89 |
| Odonata | −0.05 | 0.30 | −0.63 | 0.53 | 0.87 | 87 |
| Fish | −0.89 | 0.56 | −1.98 | 0.20 | 0.86 | 86 |
| Coleoptera | −0.04 | 0.35 | −0.73 | 0.65 | 0.84 | 84 |
| Diptera | 0.70 | 0.49 | −0.26 | 1.66 | 0.84 | 84 |
| Hirudinea (Presence) | −0.41 | 0.43 | −1.25 | 0.44 | 0.69 | 69 |
| **Hydracarina** | **−1.42** | **0.55** | **−2.50** | **−0.33** | **0.58** | **58** |
| Decapoda (Presence) | 1.76 | 1.23 | −0.66 | 4.17 | 0.53 | 53 |
| Hemiptera | −0.07 | 0.33 | −0.72 | 0.57 | 0.22 | 22 |
| Oligochaeta (Presence) | −0.02 | 0.48 | −0.96 | 0.92 | 0.14 | 14 |
| **Ephemeroptera** | **−0.84** | **0.25** | **−1.33** | **−0.35** | **0.13** | **13** |

Variables within each category are ordered by their relative importance. Variables with their 95% CI with the same sign are represented in bold. Rare aquatic taxa are introduced in the model as Presence/Absence variables, while relative abundance is used for more abundant taxa. Results for lentic and lotic ecosystems represent the decrease in *MU* respective to stagnant ecosystems.

(*Table 2*). Finally, the orders Oligochaeta, Odonata, and Hydracarina had a large importance in the final models ($w_i > 0.80$), but their effect on *MU* presence was not significant. Results for Bankim revealed a significant positive effect in all models for Shannon's diversity index ($b = 4.29$; 95% CI [1.93, 6.66], $w_i = 1$) and nearly significant for total abundance ($b = 0.86$; 95% CI [−0.06, 1.79], $w_i = 1$). In addition, most taxonomic orders appeared in the final models for this region, but their effect was unclear (*Table 3*). The estimates for taxonomic groups with a relative importance over 0.8

(Gastropoda, Anura, Trichoptera, Odonata, Fish, Coleoptera, and Diptera) had considerable uncertainty, with upper and lower CIs of opposite sign. The only two orders with a significant CI were Hydracarina ($b$ = −1.42, 95% CI [−2.50, −0.33]) and Ephemeroptera (−0.84; 95% CI [−1.33, −0.35]), but their relative importance was relatively low ($w_i$ = 0.58 and $w_i$ = 0.13, respectively).

## Discussion

Understanding how environmental factors influence host–pathogen interactions in complex natural systems, where multiple feedbacks between biotic and abiotic factors take place, is especially important in the context of multi-host and environmentally persistent pathogens. In this study, we identify abiotic and biotic drivers that may promote or block *MU* transmission in aquatic communities in two climatically distinct regions of Cameroon through a comprehensive multi-model selection procedure. In Akonolinga, we show that *MU* follows seasonal dynamics and is mainly present in waters with higher *pH* and within low abundance communities, notably those with low abundance of Gastropoda and other orders such as Decapoda, Hemiptera, or Anura. In Bankim, we show that *MU* is most prevalent in stagnant ecosystems and those with low water flow, with highly diverse (and abundant) communities.

A seasonal effect for *MU* presence in Akonolinga remains in our final models after accounting for abiotic and biotic parameters in the water bodies, which also vary seasonally. This suggests that seasonal fluctuations in *MU* presence might be directly related to climatic pressures (*Figure 1*). Indeed, while the seasonal effect is not directly linked to rainfall dynamics (Pearson's correlation test, p = 0.45), it is highly correlated with the 3-month mean rainfall accumulation in the region (Current month, plus two previous months; Pearson's correlation test, p < 0.01). As a result, we propose that the cumulative effect of rainfall over several months, increasing water levels in the environment either boosts *MU* growth or washes it off from other environmental matrices (mud, soil, plants) to aquatic ecosystems, as previously suggested from epidemiological evidence (*Morris et al., 2014*). Furthermore, given the slow growth of *MU* (*Palomino and Portaels, 1998*; *Stinear et al., 2007*), the 2-month delay between the peaks in the dynamics of rainfall and those of the seasonal effect could

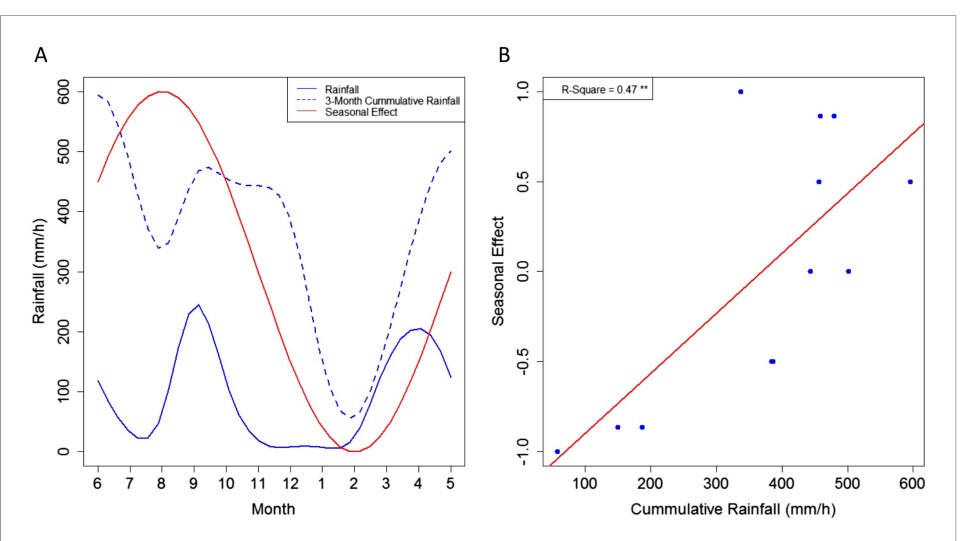

**Figure 1**. Link between the seasonal effect for *M. ulcerans* and the rainfall dynamics in Akonolinga. (**A**) Represents the monthly values for the seasonal effect (red), the mean rainfall for the period under study and the 3-month cumulative rainfall (blue). (**B**) Shows a clear linear relationship between the values of the seasonal effect and the 3-month cumulative rainfall. A graphical representation of the different seasonal effects tested can be found in *Figure 1—figure supplement 1*.

The following figure supplement is available for figure 1:

**Figure supplement 1**. Values for the different seasonal effects tested in the statistical models.

represent the time that takes *MU* to grow and/or be transmitted through the aquatic community once the suitable habitats have been created. Unfortunately, the less frequent sampling in Bankim (only 4 months instead of 12) may have prevented to capture seasonal variations appropriately, explaining the lack of associations with *MU* in this region.

Our results for Bankim support the hypothesis that, under certain circumstances, conditions in stagnant and slow-flowing (lentic) bodies of water are favourable for *MU* presence. After controlling for all the other abiotic and biotic factors, sites with stagnant waters in this area have higher *MU* positivity than those with lentic waters (slow flow), and these have in turn higher positivity than sites with lotic waters (faster flow). PCA on physico-chemical parameters of these ecosystems provides some potential explanations (*Figure 2*). Sites with stagnant or lower water flows have higher temperatures (PC1 and PC2) and most have lower oxygen (PC1) and lower *pH* (PC1), all of which seem to promote *MU* growth in experimental studies (*Portaels and Pattyn, 1982*; *Palomino and Portaels, 1998*;

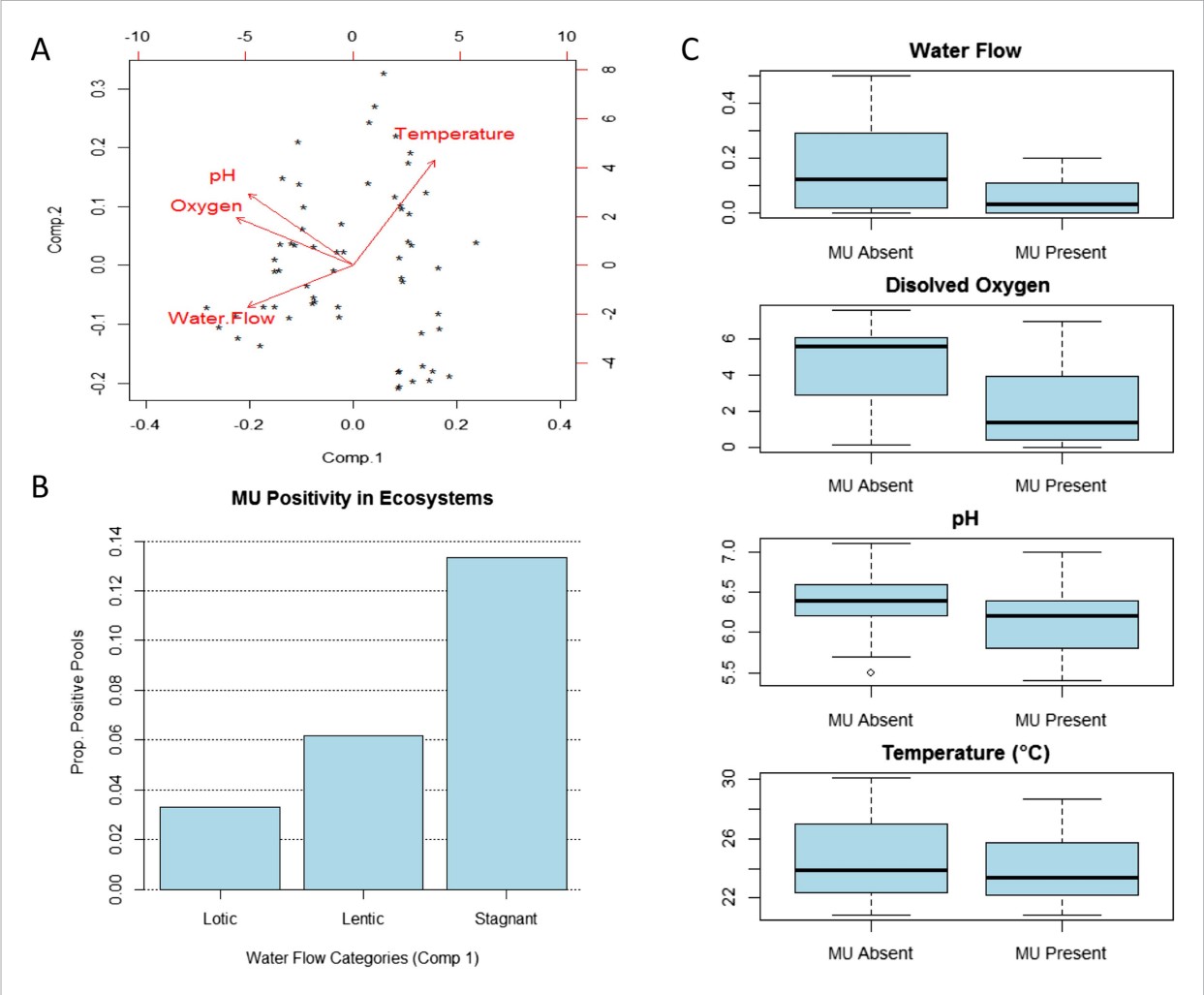

**Figure 2**. Impact of water flow on physico-chemical characteristics of the water and *M. ulcerans* prevalence in aquatic communities (Bankim). (**A**) Links between water conditions in the first two principal components obtained through principal component analysis (PCA). Comp.1, explaining more than 50% of the variation in physico-chemical conditions in Bankim, reveals that ecosystems with lower water flows have less dissolved oxygen, more acidic *pH*, and higher temperature. (**B**) *MU* positivity in each type of ecosystem as described by the first component of the PCA, which takes into account variations in all physico-chemical characteristics (each category has equal number of points and increasing values of Comp.1). Stagnant ecosystems in Bankim have higher *MU* positivity than lentic, and these have in turn higher *MU* positivity than lotic ecosystems. (**C**) Difference in values for the various water conditions in *MU* positive and *MU* negative sites in Bankim. As a result of the association of water flow with the other physico-chemical conditions, similar patterns for *MU* positivity can be observed for most abiotic conditions.

*Palomino et al., 1998*; *Dega et al., 2000*; *Marsollier et al., 2004*). Indeed, a previous field study suggested that some water characteristics may be important for the presence of mycobacteria in water and biofilms throughout the year (*Hennigan et al., 2013*).

While stagnant waters with lower *pH* contribute significantly to *MU* presence in Bankim, the results for Akonolinga show a positive association between *pH* and *MU* in this region. Significantly lower *pH* values in Akonolinga than in Bankim (t-test, p < 0.001) may be behind the disparity between the model results for each region (*Figure 3*). Indeed, *pH* range for slow-growing mycobacteria has been estimated between 5.8 and 6.5 (*Portaels and Pattyn, 1982*), which corresponds to the lower range of *pH* in Bankim, associated with stagnant waters. Because in Akonolinga, this optimal range corresponds to the upper range of values, stagnant waters with intolerably low *pH* might not meet all the optimal conditions for *MU* growth, which would explain the lack of association with these ecosystems (*Garchitorena et al., 2014*). The role of *pH* on *MU* growth in combination with other

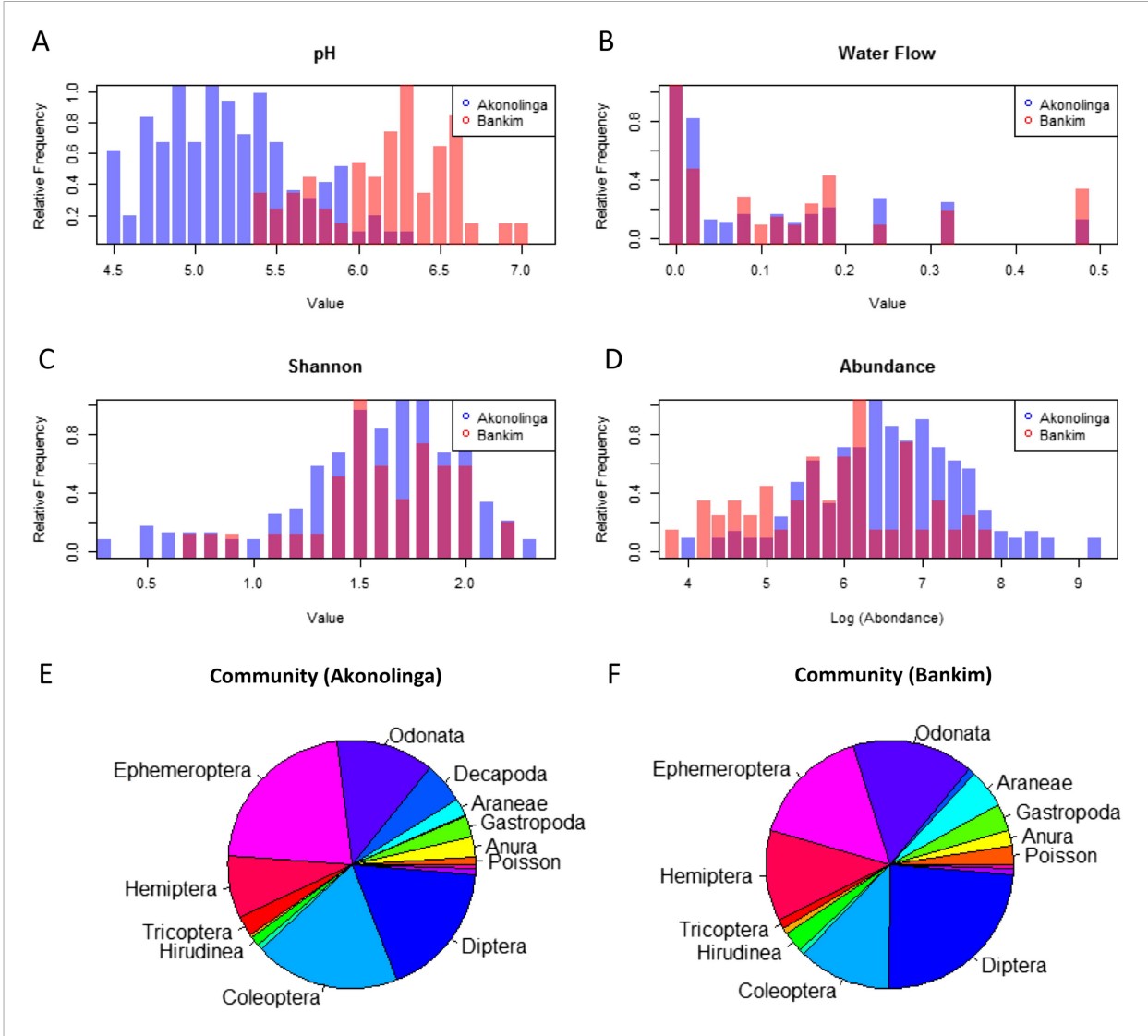

**Figure 3**. Distribution of relevant biotic and abiotic variables for Akonolinga and Bankim. For the construction of histograms (**A–D**), the relative frequency of the variable within each region is normalized by dividing each frequency by its maximum frequency. It can be noted that the distribution of *pH* is radically different for both regions, with much more acidic *pH* in aquatic environments from Akonolinga. For the community composition (**E** and **F**), the area an order has in the pie chart is proportional to the mean relative abundance of the order for all sites and months for each region. Only orders representing more than 1% of the overall community are labelled.

abiotic conditions needs to be urgently assessed, since it might be an important limiting factor in the environment.

Biotic interactions seem to have an important effect on *MU* positivity in the local aquatic communities, especially in Akonolinga. Less abundant communities are associated with a reduction of *MU* in this region, and individual taxa have an independent effect on *MU*. Higher relative abundance of aquatic snails (Gastropoda), shrimps (Decapoda), water bugs (Hemiptera), and tadpoles (Anura) is associated with reduced *MU* prevalence in the aquatic community. The protective role of aquatic snails is supported by experimental infections, where *MU* has been unable to grow within these organisms (*Marsollier and Sévérin, 2004*), but this is not the case for Hemipteran water bugs, where *MU* can grow and even colonize their salivary glands after they have fed on infected prey (*Marsollier et al., 2002*, *2005*; *Mosi et al., 2008*). Even though water bugs can host *MU* and allow its growth, they are voracious predators of aquatic organisms, and therefore, an increase in water bugs in the community may result in a decrease of infected prey available to other predators such as Coleoptera or Odonata; this could result in an overall reduction in *MU* positivity. This is an example of how considering the full breadth of factors taking place in real ecological systems can provide unexpected insights on this type of host–pathogen interactions. Furthermore, differences in community composition may partly explain the different effects of biotic factors in the two regions (*Figure 3C–F*). Total abundance in aquatic communities was significantly higher in Akonolinga (Mann–Whitney test, $p < 0.001$), and the relative abundance of more than half of the taxa included in our model was significantly different between Akonolinga and Bankim (Appendix 1, section 2).

These different distributions of biotic and abiotic factors can nevertheless yield a hypothesis to explain the contrasting results between the two regions. Indeed, two transmission routes, not competitively exclusive, may coexist for *MU* colonization of aquatic organisms, through a trophic transmission (*Roche et al., 2013a*) and/or a pathogen transmission through infection with free-living stages present in favourable aquatic environments (*Merritt et al., 2010*). In our study, community abundance has opposite effects in Akonolinga and Bankim, while water flow and *pH* suggest contrasted influence of stagnant waters in these regions. These results could suggest that the prevailing transmission modes could be different within these two environmentally distinct regions. Transmission could be mainly environmentally mediated in Bankim, since stagnant waters, through weak water flows and optimal physico-chemical conditions, are strongly associated with *MU* presence. Furthermore, a lack of association of *MU* abundance with specific taxa in addition to strong positive associations with host diversity and abundance under these favourable conditions, suggests that infection probability in these environments is density dependent, a characteristic feature of this type of transmission (*Codeço, 2001*). Conversely, in Akonolinga, trophic transmission may be expected since optimal abiotic conditions are not met in stagnant ecosystems, and host abundance has a negative impact on *MU* presence, suggesting that presence of some taxa, at least Gastropoda and Hemiptera, can limit transmission in aquatic communities. These alternative transmission routes proposed for *MU* to persist and thrive in aquatic ecosystems could partly explain why BU distribution in humans is greatly associated with stagnant ecosystems but expands over larger geographical regions (*Brou et al., 2008*; *Wagner et al., 2008*; *Jacobsen and Padgett, 2010*; *Marion et al., 2011*).

Our results demonstrate the complex interplay between abiotic and biotic factors driving the dynamics of multi-host/multi-environment diseases. By studying and comparing savannah- and rainforest-like regions, we provide a comprehensive ecological picture of *MU*, that is, a unified framework that reconciles the many contrasting findings observed during the last decade that could apply to a broader geographical area in the tropics and could help us understand the risk of BU for human populations. This study provides a new illustration of emerging infectious diseases for which further investigations looking for a 'bigger picture' are clearly needed in order to cope with the complexity of local and regional environmental situations, and different-scale processes. Judging by the number and importance of multi-host and environmentally persistent pathogens in the total number of emerging infectious diseases appeared in the last four decades such hypotheses deserve to be rigorously tested across multiple epidemiological systems and diverse local conditions. More comprehensive environmental studies in other contexts are needed to assess the generalizability of our findings.

## Materials and methods

### Environmental data collection

Data were collected as described in *Garchitorena et al. (2014)*. Briefly, between June 2012 and May 2013, periodic sampling in aquatic ecosystems was performed monthly in Akonolinga and every 3 months in Bankim, two regions in Cameroon where BU is endemic. Akonolinga health district is located in the Centre Province, where rainforest is predominant all across the region. Bankim, on the other hand, is a health district located in the Adamaoua Province, near the border with Nigeria, in a transition zone between forest and savannah. In all, 32 water sites were selected (16 in each region), including a large variety of streams, rivers, swamps, and flooded areas.

#### Aquatic macro-invertebrates and vertebrates

In each water body, four locations were chosen in areas of slow water flow and among the dominant aquatic vegetation. At each location, five sweeps were done with a metallic dip net (32 × 32 cm, 1 mm mesh size) within a surface of 1 m$^2$ and at different depth levels. All aquatic organisms collected were identified, classified with the use of taxonomic keys and a binocular microscope, and put separately into tubes with 70% ethanol.

#### Physico-chemical characteristics of water bodies

Quantitative measures of the water included turbidity (Secchi disc), *pH*, dissolved oxygen, conductivity, and temperature (Multi 3430 with SenTix 940, TetraCon 925, and FDO 925 probes, WTW, Germany). Measures with the probes were taken at 0.5 m depth and only stable values were reported. Test strips for phosphates, iron, and sulfates (Merckoquant, Germany) were used to measure specific ion concentrations near the sediment–water interface. Water flow was assessed visually by measuring the speed of a floating object over the water surface.

#### *M. ulcerans* DNA extraction, purification, and detection

Aquatic organisms from the same site and month were pooled for qPCR analysis by groups belonging to the same taxonomic group. At least six sample pools containing the most abundant taxonomic orders were analysed per site and month. Pooled individuals were all ground together and homogenized. DNA from homogenized insect tissues was purified using QIAquick 96 PCR Purification Kit (Qiagen, France). Amplification and detection were performed by quantitative PCR of the gene sequence encoding the ketoreductase B domain (KR) of the mycolactone polyketide synthase (*Rondini et al., 2003*; *Fyfe et al., 2007*) and the GenBank IS*2404* sequence (*Rondini et al., 2003*). At least 10% negative controls were included during the purification and amplification steps for each assay. Samples were considered positive only when both sequences were detected, with threshold cycle values strictly <35 cycles (see *Garchitorena et al., 2014* for details on pooling strategies and PCR analysis).

### Data analysis

#### Statistical model

The proportion of *M. ulcerans* positive sample pools at each sample collection (one site and month) was modelled using binomial regressions in GLMMs (*Zuur et al., 2009*). Since repeated samples were taken from the same sites at regular intervals during one year, we introduced the collection site as a random intercept to control for within-site correlations. In this model, we studied the effect of seasonality, physico-chemical characteristics of the water, and community composition (*Table 4*), all of which were introduced as fixed effects without interactions.

#### Multi-model selection and inference

Multi-model inference is increasingly recognised as an alternative approach to the use of null hypothesis testing (*Burnham and Anderson, 2002*; *Grueber et al., 2011*). This approach allows exploring a comprehensive set of potential models obtained as a result of multiple combinations of the explanatory variables. Instead of considering a unique final model, as is the case in classical forward, backward, or stepwise model selection procedures, with multi-model selection, it is possible to identify a set of 'top models' that can be ranked and weighted according to information criteria such as AIC. Model averaging within this set of top models provides quantitative measures of each variable's relative importance (Akaike Weights, $w_i$) and allows obtaining robust parameter estimates

**Table 4.** Description of explanatory variables from our environmental data set and their usage in the statistical model

| Variable | Min | Max | Median | Prop. zeros | Prop. NAs | | Usage |
|---|---|---|---|---|---|---|---|
| Physico-chemical parameters | | | | | | | |
| Temperature | 20.9 | 30.2 | 23 | 0 | 0 | – | Raw |
| pH | 4.5 | 7.1 | 5.5 | 0 | 0 | – | Log |
| Dissolved oxygen | 0.01 | 7.6 | 2 | 0 | 0 | – | Log |
| Conductivity | 10.2 | 110.6 | 22.7 | 0 | 0 | – | Log |
| Water flow | 0 | 0.5 | 0.03 | 0.34 | 0.01 | – | Categorical |
| Turbidity | 2 | 250 | 50 | 0.19 | 0.19 | – | Removed |
| Iron | 0 | 10 | – | 0 | 0 | – | Categorical |
| Phosphates | 0 | 250 | – | 0.07 | 0.07 | – | Removed |
| Sulphates | 0 | 600 | – | 0.22 | 0.22 | – | Removed |
| Aquatic community | | | | | | | |
| Abondance | 46 | 10,591 | 686.5 | 0 | 0 | – | Log |
| Shannon | 0.35 | 2.34 | 1.7 | 0 | 0 | – | Raw |
| Aquatic taxa (%) | | | | | | | |
| Fish | 0 | 0.32 | 0 | 0.32 | 0 | Aquatic | Log |
| Anura | 0 | 0.54 | 0 | 0.33 | 0 | Aquatic | Log |
| Gastropoda | 0 | 0.8 | 0 | 0.37 | 0 | Aquatic | Log |
| Bivalvia | 0 | 0.13 | 0 | 0.91 | 0 | Aquatic | Removed |
| Araneae | 0 | 0.3 | 0.01 | 0.01 | 0 | Terrestrial | Removed |
| Decapoda | 0 | 0.59 | 0 | 0.68 | 0 | Aquatic | Dichotomous |
| Odonata | 0 | 0.54 | 0.11 | 0.02 | 0 | Aquatic | Log |
| Ephemeroptera | 0 | 0.78 | 0.16 | 0.03 | 0 | Aquatic | Log |
| Hemiptera | 0 | 0.41 | 0.08 | 0 | 0 | Aquatic | Log |
| Trichoptera | 0 | 0.19 | 0 | 0.35 | 0 | Aquatic | Log |
| Lepidoptera | 0 | 0.12 | 0 | 0.44 | 0 | Terrestrial | Removed |
| Plecoptera | 0 | 0.01 | 0 | 0.92 | 0 | Aquatic | Removed |
| Oligochaeta | 0 | 0.18 | 0 | 0.69 | 0 | Aquatic | Dichotomous |
| Hirudinea | 0 | 0.45 | 0 | 0.58 | 0 | Aquatic | Dichotomous |
| Coleoptera | 0.01 | 0.94 | 0.1 | 0 | 0 | Aquatic | Log |
| Diptera | 0 | 0.79 | 0.15 | 0 | 0 | Aquatic | Log |
| Hydracarina | 0 | 0.11 | 0 | 0.31 | 0 | Aquatic | Log |
| Collembola | 0 | 0.06 | 0 | 0.4 | 0 | Terrestrial | Removed |
| Cladocera | 0 | 0.24 | 0 | 0.47 | 0.63 | Aquatic | Removed |

and addressing the uncertainty associated with them (*Burnham and Anderson, 2002*). This methodology can be particularly appropriate in the study of complex ecological systems, where multiple interactions take place, and the interest is in finding strong and consistent predictors of a particular outcome. In our study, variables with a relative importance ($w_i$) larger than 0.8 in these sets of best models and with consistent sign (positive or negative) within the CI were considered to have strong support as predictors of *MU* positivity (*Calcagno, 2013*).

Multi-model selection of GLMMs generated with the package 'lme4' (*Bates et al., 2013*) was performed using the package 'GLMulti' (*Calcagno, 2013*) in R statistical software (*R Development Core Team, 2011*), which uses a genetic algorithm to improve the efficiency of model selection. Within the set of the 100 best models found, only those with an AIC within 2 units difference from the

best model were considered (*Bolker, 2008*). The package 'AICcmodavg' (*Mazerolle, 2013*) was used to estimate model-averaged fixed effects, unconditional standard errors, and 95% CIs. In addition to the set of models obtained using multi-model selection procedures as described above, we used classical forward–backwards selection procedures to provide complementary information and to strengthen the results obtained in this section (Appendix 1, section 1).

## Hypotheses and use of variables

### Effect of seasonality on MU

Several studies have reported seasonal variations in *MU* positivity, which could reflect an indirect influence of climate mediated through temporal changes in abiotic conditions and abundance of aquatic organisms, or a direct influence, mediated through wash-off of *MU* to the aquatic environment or increased water availability. By including a seasonal effect in a model where temporal changes in abiotic and biotic factors are taken into account, a significant independent effect of seasonality would suggest a direct impact of climate on *MU*. Seasonality was included in the model by transforming the month of collection with *sine* and *cosine* functions with different frequencies (seasonality of 4 or 12 months), as previously described (*Stolwijk et al., 1999*; *Christiansen et al., 2012*). Furthermore, the association of these seasonal functions with observed patterns of monthly and cumulative rainfall in the region was investigated to provide a biological explanation to this potential seasonal effect.

### Effect of abiotic conditions on MU

Abiotic water conditions could have a direct effect on *MU*, allowing it to grow or persist as free-living stages, or an indirect effect, through their influence on community composition. By including them in a model that takes into account the impact of aquatic taxa on *MU*, a significant independent effect of abiotic factors would suggest that these conditions are favourable to *MU* growth or persistence. Physico-chemical characteristics of water bodies were log transformed when necessary to approximate a Gaussian distribution (*pH*, dissolved oxygen, conductivity). We also transformed water flow into a categorical variable with three levels, stagnant (0 m/s), lentic (0–0.1 m/s), and lotic (>0.1 m/s). In addition to their individual effect, since several water characteristics correlate and define specific environments, we performed a PCA (using the correlation matrix) on the most relevant physico-chemical parameters (water flow, temperature, dissolved oxygen, and *pH*), and the loadings of the three PCs were included in the analysis as explanatory variables in order to remove the co-linearity between them. Finally, variables with more than 5% missing values (turbidity, phosphates, and sulphates) were discarded in the multivariate analysis to allow for comparable AICs during model selection.

### Effect of biotic interactions on MU

The multi-host nature of *MU* implies that, through biotic interactions, individual taxa as well as community-level factors could influence *MU* prevalence. By studying these factors in combination with abiotic conditions, we can not only identify the most relevant ones but also gain insight into the contribution of the two *MU* transmission routes previously described. For instance, a positive *MU* association with community abundance or diversity in environments with favourable conditions would be suggestive of density dependent or direct transmission. On the contrary, if trophic transmission was the main transmission route, likely in environments with unfavourable conditions, *MU* could be positively or negatively associated with host taxa depending on their competency. To study community composition, we calculated total abundance and Shannon index (at the taxonomic order level) for each aquatic community. Relative abundance of each aquatic taxon (taxon abundance/total abundance) was also included. Since total and relative abundance of aquatic taxa were Poisson distributed, these were log transformed to avoid problems related to the skewness of variable distributions. Furthermore, the less abundant aquatic taxa (with more than 40% zero values) were introduced as dichotomous variables, and we removed the very rare taxa (with more than 90% zero values; essentially Plecoptera and Bivalvia). Semi-aquatic or terrestrial taxa collected during the aquatic sampling (Lepidoptera, Araneae, Collembola) were not included either, since they are not likely to play an important role in the aquatic community. Finally, the taxon Cladocera, with more than 5% missing values, was also discarded.

## Acknowledgements

We are grateful to Roger Kamgang and Joachim Ossomba at Centre Pasteur du Cameroun for their contribution to the data collection and Jérémie Babonneau and Myriam Benhalima at INSERM for their contribution to the molecular analysis of samples. Thanks to all the members of the consortium ANR-EXTRA MU (Arnaud Fontanet, Jordi Landier, Gaëtan Texier, Philippe Le Gall, Estelle Marion, Danny Lo Seen) for their support and insightful discussions. We are thankful to Gabriel Garcia Peña and Kevin Carolan for advice during data analysis and comments on the first version of the manuscript.

## Additional information

### Funding

| Funder | Grant reference | Author |
| --- | --- | --- |
| Agence Nationale de la Recherche | ANR 11 CEPL 007 04 EXTRA-MU | Andrés Garchitorena, Jean-François Guégan, Lucas Léger, Sara Eyangoh, Laurent Marsollier, Benjamin Roche |
| École des hautes études en santé publique | | Andrés Garchitorena, Jean-François Guégan |
| Institut de recherche pour le développement | JEAI AtoMyc | Andrés Garchitorena, Jean-François Guégan, Sara Eyangoh |
| Fondation Raoul Follereau | | Laurent Marsollier |
| Institut national de la santé et de la recherche médicale | Programme Inserm Avenir, AtOMycA | Laurent Marsollier |
| Region Pays de la Loire | ARMINA project | Laurent Marsollier |
| Fondation pour la Recherche Médicale | FDT20130928241 | Laurent Marsollier |

The funders had no role in study design, data collection and interpretation, or the decision to submit the work for publication.

### Author contributions

AG, Conception and design, Acquisition of data, Analysis and interpretation of data, Drafting or revising the article; J-FG, SE, Conception and design, Drafting or revising the article; LL, Analysis and interpretation of data, Drafting or revising the article; LM, Drafting or revising the article, Contributed unpublished essential data or reagents; BR, Conception and design, Analysis and interpretation of data, Drafting or revising the article

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

## Appendix 1

## Section 1: backwards–forward model selection

The multi-model selection approach was chosen for its improved capabilities when compared to standard selection approaches, since it allows accounting for and describing the uncertainty in the parameters due to model selection. As a result, this approach and its results for each region are described in the main text. In this section, we describe the methodology and results of additional analyses using a classical forward–backward selection approach. We demonstrate that this selection procedure gives comparable results to those from the multi-model selection described in the main text.

### Methodology

A combination of backward and forward procedures based on Akaike's Information Criteria (AIC) and likelihood ratio tests was used to select the final multivariate model. Firstly, all variables were tested individually as fixed effects in univariate binomial regressions with the site of collection as random effect. Those variables showing a significant effect were retained for an initial multivariate model. A backward procedure was then applied on this initial model in order to select, one by one, the variables that did not significantly improve the model (based on likelihood ratio tests) and, among these, we dropped those variables that resulted in the model with the lowest AIC. Secondly, a forward procedure was carried out by adding to this reduced model, one by one, all variables that significantly improved the model (likelihood ratio test) and, among those, those that resulted in the model with the lowest AIC. At each step, we checked whether the addition of the new variable made others insignificant, in which case, we dropped those variables from the model. The final model was obtained when no significant improvement could be achieved with the addition of new variables. Violation of model assumptions was checked in the final models. Colinearity was assessed through graphical exploration of explanatory variables, correlation tests, and variance inflation factors (VIF) in the final model. Independence was assessed by studying the spatial (correlograms) and temporal correlation (cross-correlations) of the model residuals.

### Multivariate results

Similarly to the results obtained with multi-model selection, a combination of seasonal factors, water conditions, and community composition remained in the final models for Akonolinga and Bankim (*Appendix table 1*). Firstly, the effect of seasonality remained for Akonolinga after accounting for water parameters and biological factors. Secondly, regarding water conditions, *pH* had a significant positive effect on *MU* presence in Akonolinga, whereas in Bankim the negative effect of water flow was evident, with lotic ecosystems being the least positive, followed by lentic ones, which had significantly lower positivity than stagnant ecosystems. Finally, the effect of aquatic communities through total abundance was significant and inversely correlated with *MU* in Akonolinga, while in Bankim Shannon's diversity showed a positive correlation with *MU* presence. Individual taxa that remained significant and inversely correlated with *MU* were Gastropoda, Decapoda, Hemiptera and Anura for Akonolinga; and Ephemeroptera, fish, Diptera and Anura for Bankim. The only taxon positively correlated with *MU* in the final model was Hirudinea in Akonolinga and Coleoptera for Bankim. In summary, the results obtained from multi-model selection and classical model selection procedures were qualitatively and quantitatively similar.

**Appendix table 1**. Results of multivariate analyses for Akonolinga (12 months of sampling) and Bankim (4 months of sampling)

| Variable | Akonolinga (n = 183) | | | Bankim (n = 61) | | |
|---|---|---|---|---|---|---|
| | Effect | Std. error | p-value | Effect | Std. error | p-value |
| Model AIC | 400.7 | – | – | 182.7 | – | – |
| Variance of random effect | 0.20 | – | – | 1.77 | – | – |
| (Intercept) | −12.56 | 4.40 | <0.001 | −7.40 | 1.97 | <0.001 |
| Seasonality | | | | | | |
| Sine(2pi*Month/12) | 0.34 | 0.14 | 0.02 | – | – | – |
| Sine(2pi*Month/4) | – | – | – | – | – | – |
| Cos(2pi*Month/12) | – | – | – | – | – | – |
| Cos(2pi*Month/4) | – | – | – | – | – | – |
| Physico-chemical parameters | | | | | | |
| Temperature | – | – | – | – | – | – |
| pH | 8.63 | 2.44 | <0.001 | – | – | – |
| Dissolved oxygen | – | – | – | – | – | – |
| Conductivity | – | – | – | – | – | – |
| Iron | – | – | – | – | – | – |
| Water flow (Lentic) | – | – | – | −2.10 | 0.47 | <0.001 |
| Water flow (Lotic) | – | – | – | −3.18 | 0.69 | <0.001 |
| Physico-chemical parameters (PCA) | | | | | | |
| PC1 | – | – | – | – | – | – |
| PC2 | – | – | – | – | – | – |
| PC3 | – | – | – | – | – | – |
| Community | | | | | | |
| Abundance | −0.64 | 0.17 | <0.001 | – | – | – |
| Shannon | – | – | – | 4.16 | 0.97 | <0.001 |
| Orders (%) | | | | | | |
| Fish | – | – | – | −1.62 | 0.35 | <0.001 |
| Anura | −0.34 | 0.14 | 0.02 | −0.84 | 0.32 | 0.01 |
| Gastropoda | −0.64 | 0.16 | <0.001 | – | – | – |
| Decapoda (presence) | −1.37 | 0.37 | <0.001 | – | – | – |
| Odonata | – | – | – | – | – | – |
| Ephemeroptera | – | – | – | −0.94 | 0.21 | <0.001 |
| Hemiptera | −0.47 | 0.20 | 0.02 | – | – | – |
| Tricoptera | – | – | – | – | – | – |
| Oligochaeta (presence) | – | – | – | – | – | – |
| Hirudinea (presence) | 0.59 | 0.23 | 0.01 | – | – | – |
| Coleoptera | – | – | – | – | – | – |
| Diptera | – | – | – | 1.08 | 0.36 | <0.001 |
| Hydracarine | – | – | – | −1.58 | 0.49 | <0.001 |

The models used are Binomial regressions with random effect site, selected with forward–backwards procedure (see section 1 for details).

# Section 2: differences in communities in Akonolinga and Bankim

In the main text, we justify the different results obtained for Akonolinga and Bankim partly based on differences between the two regions, both in terms of physico-chemical parameters and community composition (*Figure 3*). In *Appendix table 2*, we show the relative abundance for each taxonomic group in each region and test for differences on the mean relative abundance between the regions by using Mann–Whitney tests (also known as Wilcoxon rank-sum tests).

**Appendix table 2**. Differences in community composition between Akonolinga and Bankim

| Taxonomic group | Relative abundance (%) | | | | Mann–Whitney test |
| | Akonolinga | | Bankim | | |
| | Mean | SD | Mean | SD | p-value |
| --- | --- | --- | --- | --- | --- |
| Fish | 1.04 | 2.20 | 2.39 | 4.93 | <0.001 |
| Anura | 2.56 | 6.62 | 2.08 | 6.73 | 0.009 |
| Gastropoda | 2.70 | 7.76 | 3.41 | 9.33 | 0.449 |
| Bivalvia | 0.19 | 1.08 | 0.00 | 0.03 | 0.016 |
| Decapoda | 5.36 | 12.37 | 0.90 | 2.78 | 0.010 |
| Odonata | 12.72 | 9.96 | 15.79 | 15.03 | 0.616 |
| Ephemeroptera | 21.77 | 17.22 | 15.56 | 14.74 | 0.010 |
| Hemiptera | 8.15 | 5.85 | 11.84 | 8.55 | <0.001 |
| Tricoptera | 2.62 | 4.20 | 1.23 | 2.23 | 0.011 |
| Hirudinea | 1.29 | 4.76 | 2.63 | 7.31 | 0.005 |
| Oligochaeta | 0.64 | 1.94 | 0.67 | 2.11 | 0.291 |
| Coleoptera | 18.98 | 20.54 | 11.99 | 11.37 | 0.023 |
| Diptera | 17.58 | 14.23 | 23.69 | 16.25 | 0.004 |
| Hydracarine | 0.83 | 1.38 | 0.90 | 1.77 | 0.171 |

For each taxon included in the statistical model, the mean and standard deviation (SD) of the relative abundance (%) for each region are given, along with the p-value of a Mann–Whitney test comparing the mean relative abundance in the two regions.

