## [Decision Letter]

Thank you for submitting your work entitled “*Mycobacterium ulcerans* dynamics in aquatic ecosystems are driven by a complex interplay of abiotic and biotic factors” for peer review at *eLife*. Your submission has been favorably evaluated by Ian Baldwin (Senior editor) and three reviewers, one of whom is a member of our Board of Reviewing editors. The following individuals responsible for the peer review of your submission have agreed to reveal their identity: Quarraisha Abdool Karim (Reviewing editor); Parviez Hosseini and Sourya Shresta (peer reviewers).

The reviewers have discussed the reviews with one another and the Reviewing editor has drafted this decision to help you prepare a revised submission.

Existing large environmental datasets on *Mycobacterium ulcerans* (*MU*) from two climatically distinct regions of Cameroon were analysed using a comprehensive multi-model selection procedure to characterize abiotic and biotic factors driving the dynamics of this pathogen. *M. ulcerans* is an environmentally persistent microorganism associated with freshwater ecosystems and present in a large variety of aquatic hosts. *MU* is the agent responsible for Buruli ulcer (BU), a devastating skin disease with great health and socio-economic consequences in tropical and subtropical countries. Regional differences in transmission dynamics is influenced by seasonality, pH, rate of water flow, density of biotic and abiotic systems and influence the emergence and persistence of *M. ulcerans*. Additional studies are needed to validate these findings. The approach of a mix of classical model selection procedures with cutting-edge new selection methods to identify and quantify the most important predictors of *MU* dynamics is novel and could be used to study other pathogen transmission dynamics.

Essential revisions:

1) As this is a complex manuscript with many outcomes kindly provide a key message(s) for the reader.

2) The manuscript tests multiple hypotheses – structure the Methods and Results section so that approach and results for each hypothesis being tested can be tracked easily by the reader.

3) The statistical approach is novel and an explanation or rationale for the choice of methods will be useful to help readers understand the decision making processes better.

Reviewer #1:

Host-pathogen interactions are typically complex and multifactorial. It is however important to unravel the multiple pathways and relative contributions of each to informing, customising and targeting interventions. The authors took advantage of a combination of large environmental datasets on *Mycobacterium ulcerans* (*MU*) to characterize abiotic and biotic factors driving the dynamics of this pathogen in two different regions of Cameroon. *M. ulcerans* is an environmentally persistent microorganism associated to freshwater ecosystems and present in a large variety of aquatic hosts and from a public health perspective, it is the agent responsible for Buruli ulcer (BU), a devastating skin disease with great health and socio-economic consequences in tropical and subtropical countries. They were able to identify abiotic and biotic drivers that may promote or block MU transmission in aquatic communities in two climatically distinct regions of Cameroon through a comprehensive multi-model selection procedure. They demonstrate regional differences in transmission dynamics that is influenced by seasonality, pH, rate of water flow, density of biotic and abiotic systems thereby advancing our understanding of underlying ecological mechanisms driving the emergence and persistence of *M. ulcerans*. These findings are by no means definitive and would require additional studies to validate these findings. Notwithstanding this limitation in the context of an upsurge in the emergence and re-emergence of multiple infectious diseases identifying main drivers of host–parasite interactions particularly those associated with pathology and have public health implications these types of endeavours are critically important. The findings are novel and the approach used is a mix of classical model selection procedures with cutting-edge new selection methods to identify and quantify the most important predictors of *MU* dynamics. The authors are very careful and considered in their discussion of the implications of separating out the contributions of biotic and abiotic factors for host/parasite interactions and the importance of rigorously analysing the underlying drivers of transmission dynamics mediated through complex and diverse processes.

Reviewer #2:

While in general I find this a very clear manuscript, with well-conceived and executed science, and very well described, clear, and reasonable statistical methods, I feel the paper does lack a strong take home message.

Something that could help here would be advancing the comparative rationale between the two major locations (Akonolinga and Bankim) and setting that up in the Introduction. The results seem quite different between the two sites, and there seem to be good biological reasons for this, but we don't really find that out until the Discussion.

Another question is how can these results from two specific sites be understood to generalize to a broader area, and help provide a broader understanding of *MU* risk and/or mitigation strategies. Again, I think if the comparison was more strongly setup in the Introduction, this might help.

Lastly, I think the authors really need to decide what the take home message is, and make sure they close the loop on this. Right now this is a bit of a muddled message about abiotic and biotic factors mattering depending on location, and while likely true, that message doesn't provide any insight or predictive value.

There seems to be a more specific conclusion that a certain *pH* range, which was important in the lab, is also found to be important in the field. Even in the specifics, the conclusions are a bit muddled with hydrologic regime and community being important, but not in a cross site way. Again some sort of comparative setup that helps create some generalizability would be useful here.

Additional data files and statistical comments:

I found the rigor and extent of the statistical information supplied to be commendably high.

Reviewer #3:

I find this to be an interesting and substantive work that attempts to better understand the roles of climatic factors as well as other biotic factors

in the persistence and transmission of *Mycobacterium ulcerans* (*MU*).

The paper uses extensive data collected in two endemic areas in Cameroon (Akonolinga and Bankim), and a series of statistical/data analysis tools to

explore these questions.

Although the data are described in their previous work, this paper focuses on using a suite of statistical tools to explore the role of environmental and biotic factors: a clear strength of the paper. Through the use of these methods in two different locations, the authors uncover interestingly different dynamics in the two sites:

In Akonolinga, TU persisted in less abundant aquatic communities, and showed seasonal pattern, whereas in Bankin, TU persisted in diverse and abundant communities, and showed no seasonal pattern. This has enabled the authors to propose several interesting hypotheses: The role of biotic interactions, *pH* level, and two-transmission routes.

My main comments concern making the paper more accessible and compelling.

1) The paper could be made more accessible by motivating the use of each of the statistical methods used in the paper. In particular, before carrying out a test, it would useful for the readers to have an expectation of what question a particular method is testing. The paper uses many of them, making it somewhat confusing to follow results from all of them. It would also be useful to have a better sense of what value a specific test is bringing in the context of the questions.

2) The authors propose several hypotheses that could explain the patterns observed in the data, particularly the difference seen in the two sites. These are all compelling, but somewhat difficult to follow in the Discussion. This is particularly true for the hypothesis with two transmission routes, and the hypothesis concerning the biotic interaction. I would urge the authors to organize this better, and discuss whether the hypotheses are mutually exclusive, or whether are inter-related.

Overall, I find the paper rich (in terms of the use of data and methods), rigorous (in terms of different use of statistical methods), and compelling/interesting (particularly, with regards to interesting hypotheses the authors were able to pose).

---

## [Author Response]

1) As this is a complex manuscript with many outcomes kindly provide a key message(s) for the reader.

*[…] There seems to be a more specific conclusion that a certain* pH *range, which was important in the lab, is also found to be important in the field. Even in the specifics, the conclusions are a bit muddled with hydrologic regime and community being important, but not in a cross site way. Again some sort of comparative setup that helps create some generalizability would be useful here*. *[…]*

*The authors propose several hypotheses that could explain the patterns observed in the data, particularly the difference seen in the two sites. These are all compelling, but somewhat difficult to follow in the Discussion. This is particularly true for the hypothesis with two transmission routes, and the hypothesis concerning the biotic interaction. I would urge the authors to organize this better, and discuss whether the hypotheses are mutually exclusive, or whether are inter-related*.

We agree with the referees that the message of the study was indeed buried in the very end of the Discussion in the previous version and we failed to provide a consistent message throughout the paper. Following the suggestion of the reviewer #2, we have better explained the rationale behind the potential transmission routes, and used the regional differences between the two regions from the Introduction section to illustrate the potential contributions of each transmission route in each region depending on abiotic conditions:

“A direct transmission of *MU*, driven mostly by abiotic factors […] may have profound implications for understanding BU risk to human populations.”

The manuscript has therefore been organized around the following take home message: we identify a set of abiotic conditions that could be optimal for *MU* growth and could imply a direct transmission to aquatic organisms in stagnant waters; and in the absence of these, notably because of too acidic *pH* in rainforest swamps, biotic interactions still allow *MU* to persist through trophic transmission.

The Discussion has also been modified accordingly, explaining better the implications of our results for these two transmission routes and better organizing the flow of the Discussion to separate the sections around abiotic and biotic factors, instead of discussing them together in the same paragraphs. We have also linked the key findings of the study to the potential implications in terms of risk for human populations and make an effort to improve generalizability of our results to similar geographical regions:

“Transmission could be mainly environmentally-mediated in Bankim […] and could help us understand the risk of BU for human populations.”

*2) The manuscript tests multiple hypotheses – structure the Methods and Results section so that approach and results for each hypothesis being tested can be tracked easily by the reader*. *[…]*

*The paper could be made more accessible by motivating the use of each of the statistical methods used in the paper. In particular, before carrying out a test, it would useful for the readers to have an expectation of what question a particular method is testing. The paper uses many of them, making it somewhat confusing to follow results from all of them. It would also be useful to have a better sense of what value a specific test is bringing in the context of the questions*.

We apologize for the confusion in the organization of the previous version of the manuscript, which made it difficult to follow. We have restructured the Results section to explain separately the three groups of factors (seasonal, abiotic, and biotic), instead of separating them by region. We also provide a short background of what is being tested in each case before explaining the results obtained. In addition, we have expanded the Materials and methods section to explain the hypothesis being tested for each of these three groups of factors and the expected results before explaining how they were tested in the models. This can now be found as a separate part of the data analysis subsection called “Hypotheses and use of variables”.

Materials and methods, Results and Discussion sections now follow the same logical order (seasonality, then abiotic and then biotic factors), which we hope will improve the clarity for the reader and will make it easier to track hypotheses, tests, results and implications for each of these factors through the different sections of the manuscript.

*3) The statistical approach is novel and an explanation or rationale for the choice of methods will be useful to help readers understand the decision making processes better*.

We thank the referees for allowing us to better explain the novel methodological approach used and the rationale that brought us to use it. Indeed, we feel that multi-model inference could be an appropriate alternative to overcome some of the limitations of classical selection approaches, especially when studying complex ecological systems. We have expanded the explanation of this method and its advantages in the corresponding part of the Materials and methods section, multi-model selection and inference:

“Multi-model inference is increasingly recognised as an alternative approach […] the interest is in finding strong and consistent predictors of a particular outcome.”

In addition, we have added a short summary of the approach rationale in the last paragraph of the Introduction section in order to help the reader understand the decision for the choice of method from the beginning of the article:

“[…] We use cutting-edge multi-model selection procedures and information theory to identify and quantify the most important predictors of *MU* dynamics, using a genetic algorithm to screen multiple models from all potential combinations of explanatory variables and making inference from a set of weighted best performing models. In addition, we back the results of this novel approach, which deals with the uncertainty associated with model selection, by comparing them to those obtained by classical model selection procedures.”